# Influence mechanism of Internet use on the physical and mental health of the Chinese elderly—Based on Chinese General Social Survey

**Peng Hou**[1,2]*

**1** School of Humanities and Social Science, Xi'an Jiaotong University, Xi'an, China, **2** Department of P.E, Xidian University, Xi'an, China

* houpeng@xidian.edu.cn

**Data Availability Statement:** All relevant data are within the manuscript and its Supporting Information files.

**Funding:** The author(s) received no specific funding for this work.

## Abstract

Based on Chinese General Social Survey data (CGSS 2021), binary logistic regression and stepwise regression were used to explore how Internet use improves the physical and mental health of elderly people and its influence mechanisms. The research found that Internet use has a positive and significant impact on the physical and mental health of the Chinese elderly, and the results are robust with variable replacement and model replacement tests. In its influence mechanism, it found that Internet use promotes the physical and mental health of elderly people through physical exercise, social interaction, and learning frequency, which have a partial mediating effect. The effectiveness of the Internet use in promoting physical and mental health of the Chinese elderly through learning frequency is higher than physical exercise and social interaction, highlighting the importance of continuous learning for the Chinese elderly in the digital age. At the same time, Internet use has an unequal influence on the physical and mental health of the Chinese elderly, and has a greater influence on the mental health of the elderly with higher socio-economic status. Therefore, the research proposes the following three suggestions. First, improve the popularity of Internet use among the Chinese elderly. Second, accelerate the development of Internet application products suitable for the Chinese elderly. Third, provide Internet education for different regions elderly groups, and implement targeted assistance for elderly people with poor socio-economic status.

## 1 Introduction

Now, China is rapidly advancing towards a medium aging society and the degree of population aging is becoming extremely intense [1]. By the end of 2023, the number of elderly people (above 60 years old) account for 21.1% of the population in China. According to the 2021 Statistical Bulletin on the Development of China's Health Industry, it is indicated that average life expectancy in China has reached 78.2 years. Moreover, more than 75% of people over 60 years

**Competing interests:** The authors have declared that no competing interests exist.

old suffer from one or more chronic diseases. Meanwhile, as of December 2022, China's Internet users have exceeded 1.067 billion, including 153 million elderly Internet users. Some researches and reports point out that the Chinese elderly are "lonely" on the Internet—almost online 24 hours a day, resulting in serious physical and mental health problems among the elderly [2].

The digital era is having an increasingly profound impact influence on the physical and mental health of the Chinese elderly. The combination of an aging population and digital society has given rise to the widespread concern of the "silver digital divide" in society. Some studies suggested that excessive Internet use can lead to Internet addiction [3]. On the other hand, the Internet use of the elderly have becoming increasingly diverse, shaping a more diversified cognition to a certain extent. Under the promotion of the digital age, social participation has been carried out, enhancing the accessibility of health care services for the elderly. For example, such as Tiktok live sports teaching, "WeChat sports" race and other online platform activities [4, 5]. All these activities have activated the subjective initiative of some elderly people to pay attention to physical and mental health while the Internet is aging [6]. How can we use the characteristics of the current digital society era to promote the re-socialization of the elderly with the use of the Internet, actively respond to China's aging population and making the promotion of healthy aging crucial for the physical and mental health of the elderly [7].

There is no consistent conclusion on this issue in existing literature. Part of the research results support the digital empowerment theory. It is believed that in the digital era, the Chinese elderly broaden information channels, enlarge social capital, learns and understands sports knowledge through the use of the Internet, so as to promote physical exercise and improve physical and mental health [8]. Another part of the research supports the theory of time substitution, suggesting that Internet use in the digital age will curtail the social time of the Chinese elderly, reduce the frequency of learning and physical exercise, and thus affect their physical and mental health [9, 10]. Although these two views have their own evidence, they have a common flaw: the assumption of group homogeneity in theory. That is the influence of Internet use on physical exercise participation of the elderly group is consistent regardless of whether it is positive or negative, while ignoring the differences in socio-economic background within the elderly group. According to the "Fundamental Theory", the socio-economic status factor is the basic reason that affects people's health inequality [11, 12]. Therefore, to clearly reveal the influence of Internet use characteristics of the elderly on their physical exercise behavior, we must fully consider this key factor.

Health serves as the cornerstone for enhancing life expectancy, improving quality of life, and achieving healthy aging [13]. Compared with developed countries, the Chinese elderly started using the Internet at a later stage. Furthermore, the focus on how Internet use impacts both the physical and mental health of the Chinese elderly is still not comprehensive enough, and its influence mechanism and heterogeneity warrant further research. Based on above, the article will use the data from the 2021 Chinese General Social Survey (CGSS) to establish a binary logistic regression model, combining the changing laws of the digital age, close to the real situation, explore the effect of Internet use on the physical and mental health of the elderly, with a view to further improving the physical and mental health of the elderly in China, and also promoting active aging and promoting the construction of "Healthy China 2030", which holds significant practical importance.

## 2 Literature review

Internet use refers to various forms of activities conducted by using Internet platforms and technologies, including social networking, online shopping, and gaming, developed in the

digital age and represent an integral part of life [3, 14, 15]. In the context of the rise of the digital society, Internet use and physical and mental health mainly involve two impact directions: negative impact and positive impact. Some studies believe that excessive use or even addiction to the Internet will damage personal physical and mental health, and Internet addiction is also considered a psychological disease [3, 16]. Another part of the research believes that the use of the Internet can improve the level of physical and mental health. Internet use not only improves the availability of health information, and promotes the level of physical and mental health through wide dissemination of media, but also broadens the social network, providing necessary social emotional support and social tools support [17].

On this basis, many scholars have also conducted in-depth discussions on the relationship between the Internet and the physical and mental health of the elderly, of which the following three views are highly representative. The first is the positive theory of the Internet promoting physical and mental health, which believes that the elderly who use the Internet perform better in terms of mental health and physical health, and is considered an effective tool for improving health status [18–20]. For example, some scholars have found the importance of Internet use in mitigating stress among older adults experiencing resource loss and gain, emphasizing the potential of digital interventions to promote mental health in this population [21]. Other scholars believe that the elderly view the Internet as a communication platform for expressing emotions and maintaining social relationships [22]. The Internet has established novel social connections, expanded the external social networks of older individuals and created a new social space for the elderly not limited by physical boundaries. Furthermore, the Internet has strengthened intergenerational family bonds, partially fulfilling the expectations of intimacy that older individuals have toward their children [23]. In summary, the Internet significantly enhancing their overall physical and mental health [24]. The second view is that the Internet affects health negatively [25, 26]. Some scholars found that overuse of the Internet increases the risk of Internet addiction [27], such as the elderly spend a lot of daily time on the Internet, taking up the time for exchanging family and friendship [28, 29], and their dependence on online social media will increase the health risk of Internet users [19]. Other research also suggests that excessive Internet use can lead to reduced rest time, which may harm the health of middle-aged and older adults [30]. The third view is the indirect influence theory of the Internet, which believes that the Internet, as a medium, cannot directly affect physical and mental health, but has an influence on physical and mental health through multiple mechanisms [31]. From the perspective of the interpersonal emotion, some scholars believe that the use of the Internet promotes the mental health of the elderly through social activities [32]. From the perspective of information acquisition, the elderly acquire and learn health knowledge by using the Internet to strengthen disease prevention or participate in online physical exercise and other health activities to improve their physical and mental health [33, 34]. At present, most viewpoints support the positive theory that the Internet promotes health, while the negative theory that the Internet affects health is more reflective of health problems after excessive use or dependence on social media. When discussing the influence mechanism of Internet use on physical and mental health of the elderly, the indirect relationship between Internet use and physical and mental health should also be considered.

Based on existing research and reality, Internet use has a complex influence mechanism on the physical and mental health of the elderly. On the one hand, digital technologies also contribute to health promotion and disease prevention in an aging society [35]. Internet use can provide rich health information for the elderly, expand social circles, improve social participation, and thereby promote their physical and mental health [36]. On the other hand, excessive dependence on the Internet, network security problems and lack of practical interpersonal communication may have a negative impact on the physical and mental health of the elderly

[19, 25–30]. At the same time, previous studies on the relationship between Internet use and physical and mental health of the elderly still have deficiencies. First, most previous studies on the influence mechanism of Internet use on the elderly's physical and mental health focus on one side and lack comprehensive research and judgment. For instance, the mediating effect of physical exercise between Internet use and the physical and mental health of the elderly is only theoretically possible and has not yet been tested. The social capital of the elderly mainly focuses on improving mental health, and whether there is a mediating effect on physical health still needs to be tested. Which mediating effect is the strongest when the elderly use the Internet to promote physical and mental health. The above deficiencies exist in many studies, and the multiple mediating mechanisms between Internet use and physical and mental health of the elderly still need to be clarified. Secondly, the existing research results use data from earlier periods, longer time spans, and weaker representativeness. Thirdly, whether it is positive promotion theory, negative influence theory or indirect influence theory, not only many studies have failed to elaborate the influence mechanism of Internet use on the physical and mental health of the elderly, but also there are often problems such as insufficient sample representation or lack of stability of results.

Moreover, existing research indicates that social support is crucial for the physical and mental health of the elderly, which is beneficial for maintaining and promoting their physical and mental health, and also an important source of self-efficacy. Self-efficacy is one of the most significant predictors of successful adherence to exercise plans [37], and social capital is closely related to health inequality among the elderly [38]. Therefore, in order to further explore the influence mechanism of Internet use on the physical and mental health of the Chinese elderly, the research plans to carry out three types of research and further analysis based on the theoretical framework of social support theory, social capital theory and self-efficacy theory.

From the perspective of social support theory, the material, emotional, and informational support individuals receive from social networks can help them cope with challenges and pressures in life, and achieve satisfaction in their own personality [39]. Digital literacy education programs are expected to help older adults engage in online welfare services, thereby increasing their social support, self-esteem, and overall well-being [40]. For example, online fitness courses and fitness groups can get social support while meeting new friends, so as to increase their willingness to participate in physical exercise and then improve their physical and mental health. The personal attention they receive during activities can explain the antidepressant effect of exercise and promote physical health in physical exercise. So, when the Chinese elderly use the Internet, they can obtain social support in the form of physical exercise via the Internet. Therefore, this study proposes hypothesis one: physical exercise plays a mediating role in the influence of Internet use on the physical and mental health of the Chinese elderly. From the perspective of social capital theory, social networks and social relationships can provide the exchange of resources and information, thereby enabling people to obtain an equal amount of resources. In the use of the Internet, the Chinese elderly can expand their social network through the Internet, promote social interaction in the process of Internet use. In the process of social group interaction, Chinese elderly people can increase their sense of pleasure, enhance their sense of happiness, reduce their sense of loneliness, and alleviate their depression, at the same time, participating in more social activities to obtain social capital, access more health resources and information, and improve physical and mental health [41, 42]. Therefore, this study proposes hypothesis two: social interaction plays a mediating role in the influence of Internet use on the physical and mental health of the Chinese elderly. From the perspective of self-efficacy theory, completing a task can lead to positive changes in people's physical and mental state and increase self-efficacy [43]. For the Chinese elderly, the digital lifestyle enriches their learning channels and content, and continuous learning can ease their

anxiety and other negative emotions, reduce their loneliness, slow down brain decline, and in the process of continuous learning and re socialization through the use of the Internet, can enhance one's own health awareness and prevent diseases, generate self-efficacy, and maintain and promote physical and mental health [44–46]. Therefore, this study proposes hypothesis three: learning frequency plays a mediating role in the influence of Internet use on the physical and mental health of the Chinese elderly.

Similarly, the previous literature discussed the relationship between Internet use and social inequality. Some studies maintain that groups with high socio-economic status are more likely to access the Internet and enjoy its convenience [47]. On the contrary, vulnerable groups have weaker digital literacy and face information disadvantages, leading to a digital divide [48]. However, contrary research contends that the widespread use of the Internet has increased the possibility for vulnerable groups to access information resources and technology, and promoted the upward mobility of society [49]. Therefore, this study will also further test whether there is inequality in the impact of the Internet on the health of the elderly.

It can be seen that there are still research deficiencies in existing literature. Firstly, in-depth research on the Chinese elderly is still relatively scarce. More comprehensive research is needed to explore the multiple relationships between Internet use and physical and mental health. Secondly, based on China's diverse social background, it is necessary to determine whether there is inequality in the impact of Internet use on the physical and mental health of the elderly. In brief, there is still much room for research on the Internet use of the Chinese elderly and its relationship with health. More in-depth and comprehensive exploration is needed to address the current research gaps in this field, thus providing scientific basis for formulating effective public policies.

## 3 Methods

### 3.1 Data sources

The empirical data throughout this article is from the 2021 Chinese General Social Survey (CGSS) survey [36]. This survey was conducted by the China Survey and Data Center of Renmin University of China and the China Comprehensive Social Survey Project Team. The aim is to collect data on Chinese people and various aspects of Chinese society on a regular and systematic basis, summarize long-term trends in social change, explore social issues of significant theoretical and practical significance, promote the openness and sharing of domestic social science research, and provide data materials for government decision-making and international comparative research. The survey mainly includes: social demographic attributes, housing, health, migration, lifestyle, social attitudes, class identity, labor market, social security, family, etc. It is a national, large-scale, tracking survey project. The sampling method is scientifically rigorous, and has strong sample representativeness and academic authority. The CGSS 2021 survey covers a total of 28 provinces and has good timeliness. The data includes pertinent variables such as Internet use, physical and mental health, physical exercise and personal characteristics of the Chinese elderly, which are suitable for the analysis needs of this paper. Therefore, the research extracted the Chinese elderly aged 60 and above as the analysis object, and after removing missing variables and invalid answer samples, the total sample size of the data was 2772.

### 3.2 Variable measurement

The core independent variable of the article is Internet use. Regarding Internet use, the CGSS 2021 questionnaire asked the respondents, "What is your use of the following media (Internet: including mobile Internet access) in the past year?". The research mainly focuses on the use of

the Internet. For the convenience of the research, the answer not used is assigned as 0, and the answer used is assigned as 1. The dependent variable of the article is physical and mental health, specifically including physical and mental health. The variable as follows: (1) Physical health refers to the subjective evaluation of one's own body by oneself. Measure in the questionnaire by the question "What do you think is your current physical health condition?". In this section, referring to previous research classification methods, the answers to very unhealthy, relatively unhealthy, and generally are assigned a value of 0, indicating that they are not healthy. The answers to relatively healthy and very healthy are assigned a value of 1, indicating that they are healthy. (2) Psychological health refers to the subjective evaluation of one's own psychology by the individual group. The questionnaire item is "How frequently did you feel depressed or did you feel depressed in the past four weeks?" The research refers to previous classification methods, assigning values of 0 to always, often, and sometimes answers, indicating unhealthy behavior, and assigning values of 1 to rarely and never answers, indicating healthy behavior.

Mediating variables include physical exercise, social interaction, and learning frequency. With the variables as follows: the questionnaire item is "Have you often participated in sports exercise in your free time in the past year?", as the source for physical exercise data; assign the questionnaire item "Have you frequently socialized during your free time in the past year?" as the source for social interaction data source; assign the questionnaire item "Have you often studied during your free time in the past year?" as the source for learning frequency data source. The controlled variables in the article include personal characteristic variables, social characteristic variables, family characteristic variables, and social security variables of the Chinese elderly. The measurement and descriptive statistical analysis of specific variables are shown in Table 1.

### 3.3 Model settings

**(1) Logistic regression model.**  Taking the physical health and mental health of the Chinese elderly as dependent variables, binary logistic regression models were established to analyze the influence of Internet use on the physical and mental health of the Chinese elderly. The functional relationship between variables is as follows:

$$ln\left(\frac{P_i}{1 - P_i}\right) = \beta' x_i + e \tag{1}$$

**(2) Mediating effect model.**  The causal stepwise regression method is the basic method for testing mediating effect. This paper examines whether Internet use can affect the physical and mental health of the Chinese elderly through physical exercise, social interaction and learning frequency. The paper also uses the bootstrap method to test the significance of the mediating effect. Its expression is:

$$Yi = \alpha + \alpha 0\ Xi + \Sigma \alpha j\ Zij \tag{2}$$

$$Mi = \beta + \beta 0\ Xi + \Sigma \beta j\ Zij \tag{3}$$

$$Yi = \gamma + \gamma 0\ Xi + \gamma' 0 Mi + \Sigma \gamma j\ Zij \tag{4}$$

**Table 1. Variable definition and description statistics.**

| Variables | Variable Description | Mean/ percentage | Standard deviation | Variable characteristics | Sample size |
|---|---|---|---|---|---|
| Dependent variable | | | | | |
| Physical health | Unhealthy = 0, healthy = 1 | 0.375 | 0.484 | Categorical scale variables | 2772 |
| Mental Health | Unhealthy = 0, healthy = 1 | 0.635 | 0.482 | Categorical scale variables | 2772 |
| Core independent variable | | | | | |
| Internet use | Not use = 0, use = 1 | 0.389 | 0.488 | Categorical scale variables | 2772 |
| Replace independent variables | | | | | |
| Surf the Internet or not | Not = 0, yes = 1 | 0.351 | 0.477 | Categorical scale variables | 2772 |
| Control variable Personal characteristics | | | | | |
| Age | 60 to 75 years = 0, 75 years and above = 1 | 24.6% | 0.431 | Categorical scale variables | 2771 |
| Gender | Female = 0, male = 1 | 49.1% | 0.500 | Ordered variable | 2772 |
| Marriage | No spouse = 0, with spouse = 1 | 71.0% | 0.454 | Categorical scale variables | 2772 |
| Registered residence | Rural = 0, urban and other = 1 | 41.4% | 0.493 | Categorical scale variables | 2772 |
| Education | Uneducated = 0, educated = 1 | 79.4% | 0.405 | Categorical scale variables | 2757 |
| Social characteristics | | | | | |
| Labor Participation | Not participating = 0, participating = 1 | 0.263 | 0.440 | Categorical scale variables | 2772 |
| Family characteristics | | | | | |
| Economic performance | Medium or low = 0, high = 1 | 0.083 | 0.276 | Categorical scale variables | 2772 |
| Children | From 0 to 13 | 2.307 | 1.285 | Ordered variable | |
| Social insurance | | | | | |
| Basic medical insurance | Not participating = 0, participating = 11 | 93.7% | 0.243 | Categorical scale variables | 2772 |
| Basic pension insurance | Not participating = 0, participating = 1 | 80.0% | 0.400 | Categorical scale variables | 2772 |
| Mediating variables | | | | | |
| Physical exercise | never = 1, several times a year = 2, several times a month = 3, several times a week = 4, every day = 5 | 2.724 | 1.785 | Ordered variable | 2772 |
| Social interaction | never = 1, rarely = 2, sometimes = 3, frequently = 4, very frequently = 5 | 2.541 | 1.254 | Ordered variable | 2772 |
| Learning frequency | never = 1, rarely = 2, sometimes = 3, frequently = 4, very frequently = 5 | 1.737 | 1.129 | Ordered variable | 2772 |

## 4 Result

### 4.1 Influence of Internet use on physical and mental health of the Chinese elderly

**(1) The results of binary logistic regression analysis.** Fig 1 presents the results of impact analysis based on Logistic regression, the results mainly refer to the regression results after

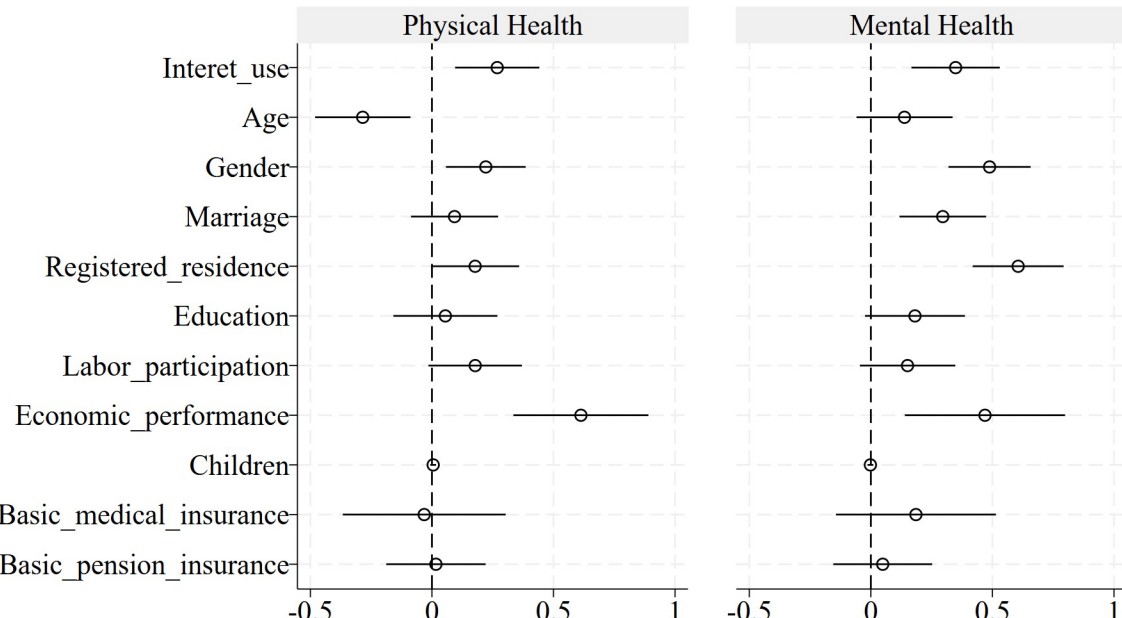

**Fig 1. Binary logistic regression results of Internet use affecting physical and mental health.** Note: The horizontal line represents the 95% confidence interval.

adding all control variables. The results show that Internet use can enhance the physical and mental health of the Chinese elderly. The Internet use has a significant positive impact on the physical health of the Chinese elderly at the 1% level, and a significant positive impact on the mental health of the Chinese elderly at the 0.1% level. Meanwhile, it can be seen that the probability of physical health and mental health of the elderly who use the Internet is 1.294 ($e^{0.258}$) times and 1.372 ($e^{0.316}$) times that of the elderly who do not use the Internet as indicated by Exp (β). Therefore, the physical and mental health of the elderly who use the Internet is better.

**(2) Robustness test.** To verify the robustness of the logistic benchmark regression results, the article intends to use two methods of variable replacement and model replacement for testing.

Firstly, robustness test is conducted by replacing the core independent variables. Take "Do you often surf the Internet in your spare time" as the replacement variable of the core independent variable "Internet use", assign the answer not to surf the Internet as 0, and the answer to surf the Internet as 1, so as to test the robustness of the model. As indicated in Table 2, Model A and Model B respectively represent the results of the robustness test of replacing independent variables on the physical and mental health of the elderly. It can be seen that replacing independent variables has a significant positive impact on the physical and mental health of the elderly. In terms of regression coefficients, the physical and mental health of elderly people who spend free time online in the replacement variables are 1.277 ($e^{0.244}$) times and 1.360 ($e^{0.308}$) times that of those who do not use the Internet. After replacing the independent variables, the coefficients and significance of Model A and Model B are basically consistent with the results of the binary logistic regression model, indicating that the logistic model has a good robustness.

Secondly, robustness test is conducted through model replacement, and linear regression models are applied to replace logistic regression models for robustness testing. As shown in Table 3, Model C and Model D respectively represent the robustness test results of the physical and mental health of the elderly based on linear regression. It can be observed that Internet use

**Table 2. Robustness test results based on variable replacement method.**

| Variables | Physical health | | | Mental health | | |
|---|---|---|---|---|---|---|
| | Model A1 | Model A2 | Model A3 | Model B1 | Model B2 | Model B3 |
| Internet use (Replace variables) | 0.392*** (0.082) | 0.276** (0.089) | 0.244** (0.090) | 0.573*** (0.086) | 0.347*** (0.095) | 0.308*** (0.095) |
| Control variable | | Controlled | Controlled | | Controlled | Controlled |
| Observations | 2772 | 2772 | 2772 | 2772 | 2772 | 2772 |
| Exp (β) | 1.481 | 1.318 | 1.277 | 1.773 | 1.414 | 1.360 |
| Constant | -0.655*** (0.050) | -0.871*** (0.115) | -0.818*** (0.206) | 0.364*** (0.048) | -0.455*** (0.111) | -0.515** (0.202) |

Note: * $p<0.05$

** $p<0.01$

*** $p<0.001$; the standard error is enclosed in parentheses.

has a significant positive impact on the physical and mental health of the elderly in China. At the same time, the coefficients and significance after replacing the linear regression model are basically consistent with those of the binary logistic regression model, which also indicates that the logistic model has a good robustness.

## 4.2 Analysis of the impact mechanism of Internet use on physical and mental health of the Chinese elderly

The article takes physical exercise, social interaction, and learning frequency of the Chinese elderly during their free time as mediating variables, applies the stepwise regression method for mediating effect analysis. The corresponding variables have been controlled in the model to analyze the influence mechanism of Internet use on the physical and mental health of the elderly. The analysis results of the mediating effects of physical exercise, social interaction, and learning frequency are shown in Table 4. Due to space limitations, the coefficients of stepwise regression and the results of the mediation test are reported together in Table 4.

Table 4 and Fig 2 show the test results of the influence mechanism of physical exercise on the physical and mental health of the Chinese elderly when using the Internet. First, Internet use has a significant impact on physical exercise (β = 0.392, <0.001), and Internet use has a significant impact on the physical and mental health of the elderly (Physical health: β = 0.258, <0.01; Mental health: β = 0.316, <0.001) indicates that the total effect is valid. Secondly, after

**Table 3. Robustness test results based on linear regression.**

| Variables | Physical health | | | Mental health | | |
|---|---|---|---|---|---|---|
| | Model C1 | Model C2 | Model C3 | Model D1 | Model D2 | Model D3 |
| Internet use | 0.093*** (0.019) | 0.068*** (0.021) | 0.060** (0.021) | 0.128*** (0.019) | 0.076*** (0.020) | 0.067*** (0.020) |
| Control variable | | Controlled | Controlled | | Controlled | Controlled |
| Observations | 2772 | 2772 | 2772 | 2772 | 2772 | 2772 |
| Constant | 0.338*** (0.012) | 0.292*** (0.026) | 0.304*** (0.046) | 0.585*** (0.012) | 0.398*** (0.025) | 0.385*** (0.045) |
| F Statistic | 24.610*** | 8.200*** | 6.889*** | 46.844*** | 28.917*** | 17.599*** |

Note: * $p<0.05$, ** $p<0.01$

*** $p<0.001$; the standard error is enclosed in parentheses.

**Table 4. Test of mediating effects of Internet use on physical and mental health.**

| | | Physical health | | | Mental health | | |
|---|---|---|---|---|---|---|---|
| | | Total effect | Direct effect | Indirect effect | Total effect | Direct effect | Indirect effect |
| Physical exercise | Coefficient | 0.258 | 0.218 | 0.040 | 0.316 | 0.293 | 0.023 |
| | Ratio to total effect | | 84.5% | 15.5% | | 92.7% | 7.3% |
| Social communication | Coefficient | 0.258 | 0.235 | 0.023 | 0.316 | 0.306 | 0.010 |
| | Ratio to total effect | | 91.1% | 8.9% | | 96.8% | 3.2% |
| Learning frequency | Coefficient | 0.258 | 0.184 | 0.074 | 0.316 | 0.275 | 0.041 |
| | Ratio to total effect | | 71.3% | 28.7% | | 87.0% | 13.0% |

adding physical exercise variables, the influence of Internet use on the physical and mental health of the elderly is still significant (Physical health: β = 0.218, <0.05; Mental health: β = 0.293, <0.01), and physical exercise has a significant impact on the physical and mental health of the elderly (Physical health: β = 0.102, <0.001; Mental health: β = 0.059, <0.05), the indirect effect values are 0.040 and 0.023, and the mediating effect accounts for 15.5% and 7.3% of the total effect, indicating that the mediating role of physical exercise in the model is partially mediated.

Table 4 and Fig 3 show the test results of the influence mechanism of social interaction on the physical and mental health of the Chinese elderly. First, Internet use has a significant impact on physical exercise ($\beta$ = 0.138, $p$<0.01), and Internet use has a significant impact on the physical and mental health of the elderly (Physical health: $\beta$ = 0.258, $p$<0.01; Mental health: $\beta$ = 0.316, $p$<0.001) which indicates that the total effect is valid. Secondly, after adding social interaction variables, the influence of Internet use on the physical and mental health of the elderly is still significant (Physical health: $\beta$ = 0.235, $p$<0.01; Mental health: $\beta$ = 0.306, $p$<0.001), and social interaction has a significant impact on the physical and mental health of the elderly (Physical health: $\beta$ = 0.167, $p$<0.001; Mental health: $\beta$ = 0.072, $p$<0.05), the indirect effect values are 0.023 and 0.010, and the mediating effect accounts for 8.9% and 3.2% of the total effect, indicating that the mediating role of social interaction in the model is partially mediated.

Table 4 and Fig 4 show the test results of the influence mechanism of learning frequency on the physical and mental health of the Chinese elderly when using the Internet. First, Internet use has a significant impact on physical exercise ($\beta$ = 0.482, $p$<0.001), and Internet use has a significant impact on the physical and mental health of the elderly (Physical health: $\beta$ = 0.258, $p$<0.01; Mental health: $\beta$ = 0.316, $p$<0.001) which indicates that the total effect is valid. Secondly, after adding the learning frequency variable, the influence of Internet use on the physical and mental health of the elderly is still significant (Physical health: $\beta$ = 0.184, $p$<0.05;

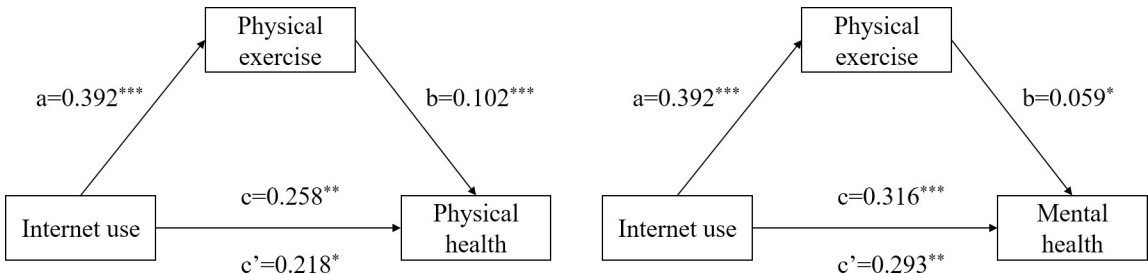

**Fig 2. Path relationship diagram of the mediating effect of physical exercise on physical health and mental health.** Note: $^*$ $p$<0.05, $^{**}$ $p$<0.01, $^{***}$ $p$<0.001.

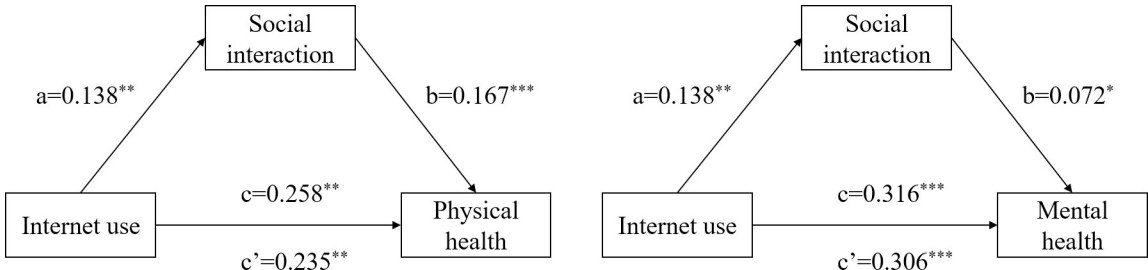

**Fig 3. Path relationship diagram of the mediating effect of social interaction on physical health and mental health.** Note: $^*$ $p<0.05$, $^{**}$ $p<0.01$, $^{***}$ $p<0.001$.

Mental health: $\beta = 0.275$, $p<0.01$), and the frequency of learning has a significant impact on the physical and mental health of the elderly (Physical health: $\beta = 0.154$, $p<0.001$; Mental health: $\beta = 0.085$, $p<0.05$), the indirect effect values are 0.074 and 0.041, and the mediating effect accounts for 28.7% and 13.0% of the total effect, indicating that the mediating effect of learning frequency in the model is partially mediated.

It can be seen that the direct effect, indirect effect and total effect of Internet use on the physical and mental health of the Chinese elderly are significant, indicating that physical exercise, social interaction, and learning frequency all have partial mediating effects, that is, Internet use can affect the physical and mental health of the Chinese elderly through physical exercise, social interaction, and learning frequency, but it can be found by comparing the proportion of indirect effects of the three mediating mechanisms, There are differences in the strength of the mediating effect among the three. Compared to physical exercise and social interaction, learning behavior has a more significant mediating effect on the physical and mental health of the Chinese elderly, while social interaction has a slightly weaker influence on the physical and mental health of the elderly.

## 4.3 The influence of Internet use on the physical and mental health of the Chinese elderly: A heterogeneity analysis based on socio-economic status

Combining with the previous content, socio-economic status is an important reason for health inequality [11, 12]. The paper uses family economic status and education as the representatives of socio-economic status, discusses the differences in the physical and mental health among the Chinese elderly with different socio-economic status in terms of Internet use. Among them, in terms of family economic status, those with an average level or below are classified as low to medium economic level, while those with an average level or above are classified as high economic level. In terms of education, those who have not received education are recorded as

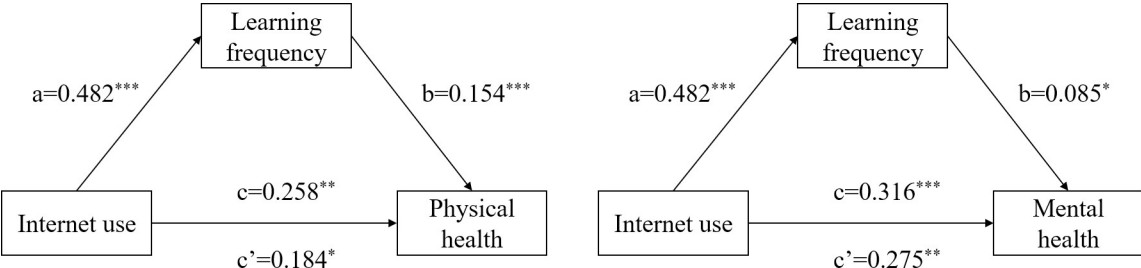

**Fig 4. Path relationship diagram of the mediating effect of learning frequency on physical health and mental health.** Note: $^*$ $p<0.05$, $^{**}$ $p<0.01$, $^{***}$ $p<0.001$.

Table 5. Heterogeneity analysis based on socioeconomic status.

| Variables | Physical health | | | | Mental health | | | |
|---|---|---|---|---|---|---|---|---|
| | Model E1 Medium-low | Model E2 High | Model E3 Un-educated | Model E4 Educated | Model F1 Medium-low | Model F2 High | Model F3 Un-educated | Model F4 Educated |
| Internet use | 0.218* (0.093) | 0.759* (0.334) | 0.438 (0.245) | 0.239* (0.096) | 0.282** (0.096) | 1.145** (0.436) | 0.335 (0.242) | 0.306** (0.102) |
| Control variable | Controlled | Controlled | Controlled | Controlled | Controlled | Controlled | Controlled | Controlled |
| Observations | 2772 | 2772 | 2772 | 2772 | 2772 | 2772 | 2772 | 2772 |
| Constant | -0.830*** (0.212) | 0.424 (1.002) | -0.452 (0.364) | -0.891*** (0.244) | -0.469* (0.206) | -0.689 (1.158) | -0.193 (0.338) | -0.485* (0.245) |

Note: * $p<0.05$

** $p<0.01$

*** $p<0.001$; the standard error is enclosed in parentheses.

uneducated, while those who have received education are recorded as educated. The specific results are detailed in Table 5.

Table 5 shows the influence of Internet use on the physical and mental health of the elderly with different family economic conditions and education. From the perspective of income, Internet use has a significant positive influence on the physical and mental health of the elderly with different family economic conditions. The promotion effect on the physical and mental health of the elderly with family economic conditions higher than average is significantly greater than that of the elderly with family economic conditions lower than average. From the perspective of education, Internet use has a significant positive influence on the physical and mental health of the educated elderly, and Internet use plays a stronger role in promoting the mental health of the educated elderly. From the above, it can be seen that the family economic status and education level of the Chinese elderly have a sustained positive influence on their physical and mental health. At the same time, Internet use has an unequal impact on the physical and mental health of the Chinese elderly, and has a greater impact on the mental health of the elderly with higher socio-economic status.

## 5 Discussion

The research value and innovation of this paper mainly lies in the introduction of physical exercise, social interaction and learning frequency as mediating variables to explore the influence of Internet use on the physical and mental health of the elderly, and the horizontal comparison of the size of the mediating effect of the three mechanisms, in order to clarify the primary and secondary relationship between Internet use and physical and mental health of Chinese elderly, and further explore its influence mechanism. At the same time, based on "Fundamental Theory" [11, 12], this paper explored the heterogeneity of different socio-economic status in the influence of Internet use on the physical and mental health of the elderly, so as to understand the inequality of Internet use on the physical and mental health of the Chinese elderly.

1. Internet use has a positive impact on the physical and mental health of the elderly in China. The physical and mental health of the elderly who use the Internet is 1.294 times and 1.372 times that of those who do not use the Internet. The elderly who use the Internet can improve their mental health more than their physical health, and the results are robust. This conclusion is consistent with the research results of the positive theory of Internet

promoting physical and mental health [18–20]. As for this result, the article believes that "Internet-addicted elderly" and "digital refugees" do exist and they affect the physical and mental health of the elderly. However, their incidence is relatively low. Moreover, with the continuous growth of experience, most elderly can maintain a relative balance between their self-awareness and control with daily life and information acquisition. In contrast, the Internet, as an important information transmission medium, has a positive impact on the physical and mental health of the elderly through the transmission of content that is beneficial to physical and mental health. At the same time, the elderly have enhanced social interaction through the use of the Internet, improved their self-confidence and sense of achievement, alleviated the anxiety and loneliness of the elderly, and obtained health information that is beneficial to themselves while sharing the achievements of the digital age, thus promoting their physical and mental health.

2. Internet use can indirectly promote the physical and mental health of the Chinese elderly through physical exercise, social interaction and learning frequency. In the current digital society, the elderly are eager to form a life mode of "doing something, enjoying something and learning something" through the Internet, so that the physical exercise, social interaction and learning frequency that occupy most of the elderly's leisure time will become more prominent in the spillover effect of the Internet. First of all, in the process of using the Internet, the elderly constantly improve their fitness awareness, understand the preventive measures for related physical diseases, and gradually change from the traditional idea of "treating the already ill" to "treating the pre ill" [50]. At the same time, in the process of physical exercise, the elderly with the same comrades work together, together with the amine metabolites or endorphins secreted during exercise, so that the use of the Internet can continue to outputs guidance paths conducive to physical and mental health for the elderly, enhance the health awareness of the elderly, thereby improving their physical and mental health level. Secondly, the elderly connect and expand the scope of their social networks through the Internet, to a certain extent, helping the elderly reshape social capital and social groups. Older people with a higher degree of similarity are more likely to exchange information about physical and mental health to enhance their enthusiasm for participating in physical exercise, and this social capital flowing in this process has accelerated the spread of health information, thus improving the physical and mental health of the elderly. Third, on the one hand, the elderly acquire physical and mental health knowledge from the Internet to improve their health literacy and prevent various diseases, on the other hand, they learn more knowledge to enhance their own value through the Internet. At the same time, in order to access and know more information, they need to be constantly familiar with the use of the Internet, learn about various terminal devices and application media, so as to further promote their re-socialization and centralization, produce a positive influence on the improvement of their physical and mental health [51].

   In addition, the research found that the use of the Internet has the largest positive influence on the physical and mental health of the Chinese elderly through learning behavior. The positive influence through physical exercise is next. And the positive impact on physical and mental health through promoting social interaction is relatively small. On the one hand, compared to physical exercise, the knowledge related to physical and mental health that elderly acquire through learning behavior is more intuitive, diverse, and efficient. On the other hand, compared with social interaction, the Chinese elderly use the Internet to exercise and learn more out of active and spontaneous behavior. And the promotion effect of physical exercise on their physical functions is more obvious in terms of visual

perception and dynamic performance. They are more likely to generate subjective initiative and emerge positive psychological emotions psychologically.

3. Through heterogeneity analysis, the research found Internet use has an unequal influence on the physical and mental health of the Chinese elderly. First of all, the educated elderly have relatively small barriers to Internet use and are more likely to acquire health knowledge and lifestyle through Internet use, thus promoting the improvement of their physical and mental health. Secondly, the Chinese elderly with higher economic status have more diverse consumption processes and choices on the Internet, and they have more opportunities to improve their physical and mental health with the help of the digital society, so their physical and mental health is better.

## 6 Conclusion

Internet use has a positive influence on the physical and mental health participation of the Chinese elderly. The higher the degree of Internet use, the higher level of the physical and mental health of the elderly. Physical exercise, social interaction and learning frequency are all intermediary mechanisms of Internet use affecting the physical and mental health of the elderly. They will indirectly promote the physical and mental health of the elderly along the path of the Internet use's influence on their physical and mental health. Among the three factors, learning frequency has the highest influence on their physical and mental health, while social interaction has the lowest influence on their physical and mental health. Internet use has an unequal influence on the physical and mental health of the Chinese elderly, and has a more obvious reinforcement effect on the elderly with high socio-economic status. The conclusion of this paper clarifies the influence of Internet use on the physical and mental health of the Chinese elderly, and provides policy enlightenment for coping with the social challenges by the aging population in China.

Based on the above conclusions, this research proposes the following three suggestions. First, increase the popularity of Internet use among the Chinese elderly. At present, the rapid development of the Internet has facilitated people's real life, but the popularity of Internet use among the Chinese elderly still needs to be improved. The government should strengthen the inclusive construction of Internet-related public services, narrow the gap in Internet penetration between urban and rural areas, improve the accessibility of Internet use for the elderly, and promote the elderly to share network resources. The social level should give full affirmation and support to the Internet use of the elderly, and give necessary guidance to the Internet use of the elderly. And encourage the elderly to actively participate in Internet use and enhance their social adaptability [52].

Second, accelerate the development of Internet application products suitable for the Chinese elderly. Although Internet products mainly focus on young people, they often ignore the needs of the elderly. With the rapid development of information technology, elderly can improve their physical and mental health through exercise, learning, and socializing, but they face difficulties in operating devices, software, and web pages. On the one hand, it is necessary to launch Internet applications suitable for the elderly as soon as possible to provide them with a better use experience. On the other hand, it is necessary to simplify the functions of terminal devices used by the elderly for Internet use and reduce the difficulty of use, so that the elderly can meet their own needs for continuous socialization while enjoying the convenient life brought by the Internet.

Third, provide Internet education for different elderly groups. Implement targeted assistance for the elderly people with poor socio-economic status. By expanding the coverage of

education and training, providing guiding videos through television and social home guidance, the elderly can master the use of the Internet and create an Internet learning field for themselves. On the one hand, enabling the elderly to achieve social participation through the Internet provides a good environment, which can effectively promote the frequency of Internet use by the elderly and expand the number of elderly using the Internet. On the other hand, strengthen the awareness of lifelong learning of the elderly, so that the use of the Internet can better play the role of information transmission, interpersonal communication and learning promotion.

## 7 Limitations and prospects

Certainly, this article still has limitations. Firstly, although the breadth and depth of the data used in the study are sufficient, the timeliness is weak. Secondly, some studies suggest that the health status of elderly is influenced by factors such as environment and genetic inheritance. Due to limitations in conditions, this manuscript was unable to fully incorporate these factors into the analysis. Third, given the complexity of the relationship between Internet use and health, future research should consider longitudinal analysis to understand better the causal relationships and temporal dynamics. Nonetheless, the research offers valuable evidence for seeking a deeper understanding of the relationship between Internet use and health. In the future, with the abundance of survey data, hope more detailed discussions can be conducted in further research.

## Supporting information

**S1 File. Data(Open with Excel).**
(XLSX)

## Acknowledgments

We thank Chinese General Social Survey for their excellent work in database design and data collection and for allowing free access to the data.

## Author Contributions

**Conceptualization:** Peng Hou.

**Data curation:** Peng Hou.

**Formal analysis:** Peng Hou.

**Writing – original draft:** Peng Hou.

**Writing – review & editing:** Peng Hou.

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
