## [Decision Letter · Decision Letter 0]

20 Aug 2024

PONE-D-24-24643Multiple impact mechanisms of Internet use on the physical and mental health of elderly people---Data Analysis Based on Chinese General Social Survey (CGSS 2021)PLOS ONE

Dear Dr. HOU,

Thank you for submitting your manuscript to PLOS ONE. After careful consideration, we feel that it has merit but does not fully meet PLOS ONE’s publication criteria as it currently stands. Therefore, we invite you to submit a revised version of the manuscript that addresses the points raised during the review process.

We look forward to receiving your revised manuscript.

Kind regards,

Mingming Li

Academic Editor

PLOS ONE

3. For studies reporting research involving human participants, PLOS ONE requires authors to confirm that this specific study was reviewed and approved by an institutional review board (ethics committee) before the study began. Please provide the specific name of the ethics committee/IRB that approved your study, or explain why you did not seek approval in this case.

6. We are unable to open your Supporting Information file [CGSS2021.sav]. Please kindly revise as necessary and re-upload.

Reviewers' comments:

Reviewer's Responses to Questions

**Comments to the Author**

1. Is the manuscript technically sound, and do the data support the conclusions?

Reviewer #1: Partly

Reviewer #2: Yes

2. Has the statistical analysis been performed appropriately and rigorously? 

Reviewer #1: Yes

Reviewer #2: Yes

3. Have the authors made all data underlying the findings in their manuscript fully available?

Reviewer #1: Yes

Reviewer #2: Yes

4. Is the manuscript presented in an intelligible fashion and written in standard English?

Reviewer #1: Yes

Reviewer #2: Yes

5. Review Comments to the Author

Reviewer #1: General Assessment

The manuscript under review investigates the impact of Internet use on the physical and mental health of elderly individuals, utilizing data from the Chinese General Social Survey and applying binary logistic regression and stepwise regression analyses. The study concludes that Internet use significantly enhances both the physical and mental well-being of elderly people. It identifies physical exercise, social interaction, and learning behavior as partial mediators in this relationship, with learning behavior having the strongest mediating effect.

Overall, while the manuscript is well-written and presents a thorough analysis, it lacks innovation in several areas. The literature review is not sufficiently comprehensive or up-to-date, and the theoretical framework and hypotheses are somewhat rigid and underdeveloped. Although three theories are mentioned, they are not explored in depth. I recommend the authors introduce more innovative models and variable selections, expand the literature review, and conduct heterogeneity analysis. The conclusion section should also provide more targeted policy recommendations.

Critical Evaluation

1. Definition and Literature Support

The manuscript's core concept, "Internet use," is not clearly defined or explained in detail. The literature cited is somewhat outdated and needs to be refreshed with more recent studies. The statement "Based on existing research and reality, Internet use has a complex impact mechanism on the physical and mental health of the elderly" lacks adequate literature support, which needs to be addressed. Moreover, the literature review and hypothesis section contain too few citations, and the discussion of the literature is not sufficiently deep. Given that the study focuses on China, it is crucial to emphasize why examining this issue in the Chinese context is important, which is not adequately reflected in the current draft.

Additionally, there are grammatical issues throughout the manuscript. For instance, the sentence “the impact mechanism of Internet use on the physical and mental health of the elderly in China still needs to be further explored and made up for the lack of existing research” is unclear and poorly structured. Another example is “Will never assign a value of 1, rarely assign a value of 2, sometimes assign a value of 3, frequently assign a value of 4, and very frequently assign a value of 5,” which is not written in a standardized manner.

2. Methodological and Innovation Concerns

The methodological approach lacks innovation. The study employs only binary logistic regression and the Mediation Effect Model, without introducing any novel methods. The selection of variables is also limited; for instance, there are no control variables related to the elderly individuals' children, which could be relevant to the analysis. Additionally, while the manuscript examines the impact of Internet use on the physical and mental health of the elderly, a topic already well-covered in the literature, it does not offer any substantial innovations. The mediation effects of physical exercise, social interaction, and learning behavior are common factors in the literature, making this study appear quite ordinary. I recommend that the authors focus on developing more innovative models and consider introducing new variables or methods to enhance the study’s contribution.

3. Conclusions and Policy recommendations

The policy recommendations provided in the manuscript are not sufficiently robust. Furthermore, the manuscript does not address the heterogeneity of the elderly population, which could be analyzed using the control variables already included in the study. To strengthen the manuscript, I suggest incorporating heterogeneity analysis and offering more specific and actionable policy recommendations in the conclusion

Areas for Improvement

To strengthen the manuscript, it is recommended that the authors:

1. Clarify and Define Key Concepts: Provide a clear and detailed definition of “Internet use” and ensure that the literature review is up-to-date and comprehensive. Expand the discussion to include more recent studies and provide a stronger theoretical grounding for the hypotheses.

2. Innovate Methodologically: Introduce novel methods or models to analyze the data. Consider including additional control variables, such as those related to the elderly individuals’ children, to offer a more comprehensive analysis. Incorporate heterogeneity analysis to better understand the diverse effects of Internet use on different subgroups within the elderly population.

3. Expand Policy Recommendations: Provide more targeted and actionable policy recommendations.

4. Improve Writing Quality: Address the grammatical issues throughout the manuscript and ensure that the writing adheres to academic standards.

Reviewer #2: The study used the CGSS data to analyze the impact of Internet using to the health of elderly people. Overall, the manuscript is well-written and the method is sound. My concerns are as follows.

(1) The Materials and Methods section is too long, in particular the “Research hypotheses” subsection. Most of the text should be discussed in the “Literature review” section while keeping the Methods section concise.

(2) The description on the variables is also unnecessarily long. Most of it is just to repeat the contents in Table 1.

(3) Why many variables are called “Dummy variable” in Table 1. They are categorical scale variables.

(4) The interpretation of eq (1) in “Model Settings” and the formula for the OR value is wrong.

(5) I would like suggest to use the forest plot of the confidence intervals of the OR values to represent the results instead of Tables.

(6) The language should be improved.

6. PLOS authors have the option to publish the peer review history of their article (what does this mean?). If published, this will include your full peer review and any attached files.

Reviewer #1: No

Reviewer #2: No

---

## [Author Response · Author response to Decision Letter 0]

7 Oct 2024

PONE-D-24-24643: Response to reviewers

Dear Editor and Reviewers:

Thank you for taking time out of your busy schedule to review the manuscript! First of all, I would like to express my sincere gratitude. Your professional review comments are very important for improving the academic research quality of this article, helping me identify shortcomings and providing constructive guidance for the revision and improvement of the paper. Thanks again! I have carefully read and analyzed all the feedback from the reviewer and editor, and have made corresponding revisions and improvements throughout the entire article. The modifications in the text have been marked in red. Revision notes, point-to-point, are given as follows:

Review-1: Response to the questions one by one

Here are the general comments from the reviewer: 

The manuscript under review investigates the impact of Internet use on the physical and mental health of elderly individuals, utilizing data from the Chinese General Social Survey and applying binary logistic regression and stepwise regression analyses. The study concludes that Internet use significantly enhances both the physical and mental well-being of elderly people. It identifies physical exercise, social interaction, and learning behavior as partial mediators in this relationship, with learning behavior having the strongest mediating effect.

Overall, while the manuscript is well-written and presents a thorough analysis, it lacks innovation in several areas. The literature review is not sufficiently comprehensive or up-to-date, and the theoretical framework and hypotheses are somewhat rigid and underdeveloped. Although three theories are mentioned, they are not explored in depth. I recommend the authors introduce more innovative models and variable selections, expand the literature review, and conduct heterogeneity analysis. The conclusion section should also provide more targeted policy recommendations.

Response: Thank you for the comments and suggestions from the reviewers. Firstly, I have added the latest literature to support the research content in the literature review section; Secondly, I provided a detailed description of the three supporting theories present in my research content; Again, new control variables were introduced and relevant models were replaced; Finally, heterogeneity analysis was conducted and more targeted policy recommendations were provided. To ensure the reviewers and editors have a clear understanding, the following is a detailed explanation of the revisions.

Critical Evaluation

1) Definition and Literature Support

The manuscript's core concept, "Internet use," is not clearly defined or explained in detail. The literature cited is somewhat outdated and needs to be refreshed with more recent studies. The statement "Based on existing research and reality, Internet use has a complex impact mechanism on the physical and mental health of the elderly" lacks adequate literature support, which needs to be addressed. Moreover, the literature review and hypothesis section contain too few citations, and the discussion of the literature is not sufficiently deep. Given that the study focuses on China, it is crucial to emphasize why examining this issue in the Chinese context is important, which is not adequately reflected in the current draft. 

Additionally, there are grammatical issues throughout the manuscript. For instance, the sentence “the impact mechanism of Internet use on the physical and mental health of the elderly in China still needs to be further explored and made up for the lack of existing research” is unclear and poorly structured. Another example is “Will never assign a value of 1, rarely assign a value of 2, sometimes assign a value of 3, frequently assign a value of 4, and very frequently assign a value of 5,” which is not written in a standardized manner.

Response: I gratefully thanks for the precious time the reviewer spent making constructive remarks. I have provided a clear and detailed definition of “Internet use” and ensure that the literature review is up-to-date and comprehensive (the first paragraph in "2 Literature review"). In addition, I expand the discussion to include recent research and provide a stronger theoretical foundation for the hypothesis(in "2 Literature review"). Mainly reflected in "1 Introduction", "2 Literature review". Finally, regarding the grammatical issues present in the manuscript, such as "the impact mechanism of Internet use on the physical and mental health of the elderly in China still needs to be further explored and made up for the shortcomings of existing research", and other relevant sentences have been refined and revised to ensure that reviewers, editors and readers have a clear understanding of the article.

2) Methodological and Innovation Concerns

The methodological approach lacks innovation. The study employs only binary logistic regression and the Mediation Effect Model, without introducing any novel methods. The selection of variables is also limited; for instance, there are no control variables related to the elderly individuals' children, which could be relevant to the analysis. Additionally, while the manuscript examines the impact of Internet use on the physical and mental health of the elderly, a topic already well-covered in the literature, it does not offer any substantial innovations. The mediation effects of physical exercise, social interaction, and learning behavior are common factors in the literature, making this study appear quite ordinary. I recommend that the authors focus on developing more innovative models and consider introducing new variables or methods to enhance the study’s contribution.

Response: Thank you very much for the valuable opinions of the reviewer and the editor! We have carefully learned to your suggestions and have specifically added the following content. Firstly, I added control variables related to the children of the Chinese elderly to provide a more comprehensive analysis (see "Table 1 Variable definition and description statistics"). Secondly, I introduced a new model to analyze the data to ensure its accuracy (see "3.3 Model settings"); Finally, the heterogeneity test and analysis part is added to better understand the different influence of Internet use on different categories of the Chinese elderly, that improve the innovation of the article, and increase the contribution of the article (see "4.3 The influence of Internet use on the physical and mental health of the Chinese elderly: a heterogeneity analysis based on socio-economic status "). 

3) Conclusions and Policy recommendations

The policy recommendations provided in the manuscript are not sufficiently robust. Furthermore, the manuscript does not address the heterogeneity of the elderly population, which could be analyzed using the control variables already included in the study. To strengthen the manuscript, I suggest incorporating heterogeneity analysis and offering more specific and actionable policy recommendations in the conclusion.

Response: Thank you for the friendly comments from the reviewer. Firstly, based on the reviewer's suggestion, I have added heterogeneity testing and analysis of elderly based on their socio-economic status (see "4.3 The influence of Internet use on the physical and mental health of the Chinese elderly: a heterogeneity analysis based on socio-economic status "). Included analysis and conclusions on heterogeneity in both the discussion and conclusion sections to better connect the framework of the article (see "5 Discussion" and "6 Conclusions" ). Secondly, based on research findings such as influence mechanisms and heterogeneity, more specific and actionable policy recommendations have been provided, mainly reflected in the "6 Conclusions" section. Finally, I added the limitations and prospects of this article to lay a foundation for future research. All specific modifications can be found in "Abstract", "2 Literature Review", "4.3 The influence of Internet use on the physical and mental health of the elderly: a heterogeneity analysis based on socio-economic status", "5 Discussion", "6 Conclusion", "Table 6 Heterogeneity Analysis Based on Socio-economic Statussection" and highlight it in red”.

Review-2: Response to the questions one by one

The study used the CGSS data to analyze the impact of Internet using to the health of elderly people. Overall, the manuscript is well-written and the method is sound. My concerns are as follows.

First of all, thank you for your meticulous guidance and inclusive suggestions on this article. Your review comments are very important for improving the academic research quality and details of this article, helping us further discover shortcomings. We carefully read and analyzed all the feedback from the reviewer, and made corresponding revisions and improvements throughout the entire document.

(1) The Materials and Methods section is too long, in particular the “Research hypotheses” subsection. Most of the text should be discussed in the “Literature review” section while keeping the Methods section concise.

Response: Thank you to the reviewer for carefully reading and promptly identifying the issues in the article. Based on the reviewer's suggestion, we have moved the "Research Hypothesis" section to "Literature Review" section for discussion, and further streamlined the content of the Methods section to make the entire Methods section more concise. The specific modifications are highlighted in red in "3 Methods".

(2) The description on the variables is also unnecessarily long. Most of it is just to repeat the contents in Table 1.

Response: Thank you for the valuable suggestions from the reviewer! The description of variables in the text does indeed overlap with the table. Based on the reviewer's suggestion, I have removed duplicate descriptions while retaining necessary variable descriptions. The variable descriptions are mainly reflected in Table 1 to maintain the simplicity of the methodology section. The specific supplementary content has been highlighted in red in "3.2 Variable measurement".

(3) Why many variables are called “Dummy variable” in Table 1. They are categorical scale variables.

Response: Thank you to the reviewer for carefully reading and providing professional revision suggestions! I have changed the "dummy variables" to "categorical scale variables". The specific modifications are highlighted in red in Table 1.

(4) The interpretation of eq (1) in “Model Settings” and the formula for the OR value is wrong.

Response: Thank you very much for the reviewer's detailed suggestions, which helped us point out the issue. I have replaced the "binary logistic regression models" and corrected the relevant description. Considering the length and fluency of the article, unnecessary descriptions unrelated to the main topic have been removed to make the structure and descriptions of the article clearer. If the editors and reviewers feel it is necessary, I can also add it back.

(5) I would like suggest to use the forest plot of the confidence intervals of the OR values to represent the results instead of Tables.

Response: Thank you very much for the valuable opinions of the reviewers! As shown in Fig 1, I have used the forest plot to display the results of the OR confidence interval, making the article fluent and the effect more aesthetically pleasing.

(6) The language should be improved.

Response: Thank you for the friendly comments from the reviewer. We deeply apologize for the language issues! To make up for this deficiency, we have once again polished and adjusted the language expression and writing throughout the article, hoping to convey the meaning more smoothly. Finally, I would like to thank the reviewer again for the constructive suggestions, which have helped me continuously improve the academic research quality of this article!

Finally, I also polished and revised the text throughout the entire paper, adjusted titles at all levels that meet the requirements of the journal, removed the highlights of the article, modified the format of the references (all highlighted in red font). To sum up, I would like to express my gratitude to the reviewers and the editors in charge for providing important revision guidance for this article, which has helped me continuously improve the academic research quality of this article. If you have any further suggestions for modification, please do not hesitate to contact me! Once again, thank you very much for your comments and suggestions!

Best wishes to you!

Author: Peng HOU

---

## [Editor Report · Decision Letter 1]

11 Oct 2024

Influence mechanism of Internet use on the physical and mental health of the Chinese elderly---Based on Chinese General Social Survey

PONE-D-24-24643R1

Dear Dr. HOU,

We’re pleased to inform you that your manuscript has been judged scientifically suitable for publication and will be formally accepted for publication once it meets all outstanding technical requirements.

Kind regards,

Mingming Li

Academic Editor

PLOS ONE
---

## [Editor Report · Acceptance letter]

31 Oct 2024

PONE-D-24-24643R1 

PLOS ONE

Dear Dr. Hou, 

I'm pleased to inform you that your manuscript has been deemed suitable for publication in PLOS ONE. Congratulations! Your manuscript is now being handed over to our production team.

Kind regards, 

on behalf of

Dr. Mingming Li 

Academic Editor

PLOS ONE